**The added value of biomarker analysis to the genesis of Plaggic Anthrosols; the identification of**
**stable fillings used for the production of plaggic manure.**
J.M. van Mourik[1*], T. V.Wagner[1], J. G. de Boer[1] and B. Jansen[1]
[1] Institute for Biodiversity and Ecosystem Dynamics, University of Amsterdam, Science Park 904, 1098 XH Amsterdam, The
Netherlands.
* Corresponding author j.m.vanmourik@uva.nl
**Abstract.**
Plaggic Anthrosols are the result of historical forms of land management in cultural landscapes on chemically
poor sandy substrates. Application of plaggic manure was responsible for the development of the plaggic
horizons of these agricultural soils. Pollen diagrams reflect aspects of the environmental development but the
interpretation of the pollen spectra is complicated due to the mix of the aeolian pollen influx of crop species
and species in the surroundings, and of pollen occurring in the used stable fillings. Pollen diagrams and
radiocarbon dates of plaggic Anthrosols suggested a development period of more than a millennium. *Calluna* is
present in almost all the pollen spectra, indicating the presence of heath in the landscape during the whole
period of soil development. Optically stimulated luminescence dating of the plaggic horizon made clear that
the deposition of plaggic covers started in the 16[th] century and accelerated in the 18[th] century. The stable
fillings, used for the production of plaggic manure and responsible for the rise of the soil surface, cannot be
identified with pollen diagrams alone. Biomarker analyses provide more evidence about the sources of stable
fillings. The oldest biomarker spectra of the plaggic horizons of three typical plaggic Anthrosols examined in this
study, were dominated by biomarkers of forests species as *Quercus* and *Betula* while the spectra of middle part
of the plaggic horizons were dominated by biomarkers of stem tissue of crop species as *Secale* and *Avena*. Only
the youngest spectra of the plaggic horizons were dominated by biomarkers of *Calluna*. This indicates that the
use of heath sods as stable filling was most likely introduced very late in the development of the Anthrosols.
Before the 19[th] century the mineral component in plaggic manure cannot be explained by the use of heath
sods. We conclude that other sources of materials, containing mineral grains must have been responsible for
the raise of the plaggic horizon.
**Key words**
Plaggic Anthrosols, plaggic manure, radiocarbon/luminescence dating, palynology, biomarkers, Netherlands.
**1. Introduction.**
Plaggic Anthrosols occur in cultural landscapes, developed on coversands. These chemical poor Late-glacial
aeolian sand deposits dominate the surface geology of an extensive area in northwestern Europe. Plaggic
Anthrosols are the characteristic soils that developed on ancient arable fields, fertilized with plaggic stable
manure. Plaggic Anthrosols have a complex genesis and are valuable records of environmental and agricultural
history (van Mourik et al., 2011).
In previous palaeopedological studies of such soil records in The Netherlands (van Mourik et al, 2011, 2012,
2013a), information was unlocked by application of pollen analysis, radiocarbon ([14]C) and Optically Stimulated
Luminescence (OSL) dating. Radiocarbon dates of soil organic carbon, extracted from humic horizons from
plaggic Anthrosols, suggested the start of sedentary agriculture between 3000 and 2000 BP but are not
indicative for the age of the plaggic sediments due to the complexity of soil organic carbon in plaggic sediments
(Mook & Streurman, 1983; van Mourik et al., 1995). It was assumed that farmers used organic sods as stable
filling, firstly dug on forest soils and later on heaths for the production of stable manure to fertilize the fields.
The mineral fraction of the sods was supposed to be responsible for the development of the plaggic horizon
and the raise of the land surface. OSL dating applied on quartz grains extracted from plaggic sediments
provides more reliable ages of the plaggic sediments. The OSL dates suggested that the rise of the plaggic
horizons started in the 16[th] century and accelerated in the 18[th] century (Bokhorst. et. al., 2005). This is rather
well in line with historical data, as presented by Spek, (2004, p 965).
The use of ectorganic matter from forest soils in the Dutch coversand area, must have been strongly reduced in
the 11[th]-13[th] century, due to commercial forest clear cuttings as recorded in archived documents (Vera, 2011).
These deforestations resulted in a regional extension of sand drifting and the managers of the heaths had to
protect their valuable ecotopes against this ´historical environmental catastrophe´ (Vera, 2011).
Heaths were already present in the Late Paleolithic landscape (Doorenbosch, 2013) and played a ceremonial
role in the society of our ancestors. People already had the knowledge to manage the heath as sustainable
grazing areas for cattle (Doorenbosch, 2013).
The use of heath for sheep grazing and other purposes as honey and oil production could continue until the
middle of the 18[th] century (Vera, 2011). In SE-Netherlands sustainable use of the heaths was promoted by
many management rules and laws (van Mourik, 1978; Veera 2011). Over the course of the 18[th] century, the
population growth resulted in an increasing food demand. In the course of the 18[th] century, the deep stable
economy was introduced and the booming demand for manure resulted in intensivation of manure production.
Farmers started with the use of heath sods as (additional) stable filling (Spek, 2004). This caused heath
degradation and initiated the second extension of sand drifting. The use of sods finished at the end of the 19[th]
century after the introduction of chemical fertilizers (Spek, 2004).
Through the combination of OSL and [14]C dating, historical records and the conventional paleoecological proxy
of fossil pollen analysis we have a good impression of the paleoecological environment and the age of such
deposits. However, it remains problematic to reconstruct the combination of crop residues and various
materials used by farmers as stable filling to produce the stable manure, together responsible for the rise of
the surface of Anthrosols. This is also hindering a detailed interpretation of the agricultural practices and shifts
therein related to the plaggic agriculture system, and specifically the timing of the onset of the intensive heath
sod driven deep stable agriculture with which plaggic Anthrosols are most commonly associated. To address
this issue, in the present study we expanded our paleoecological toolset with an adapted application of the
recently developed biomarker approach (Jansen et al., 2010). This biomarker approach consists of a
combination of analytical chemical analysis and modelling with the VERHIB model to unravel concentration
patterns of higher chain length ($C_{20}$-$C_{36}$) *n*-alkanes of higher plant origin preserved in a soil or sedimentary
archive into the (groups of) species responsible for their production (Jansen et al., 2010). The approach was
originally developed to unravel past local vegetation composition. Upon successful application in a tropical
ecosystem setting (Jansen et al., 2013), its applicability in palaeopedology was explored (van Mourik & Jansen,
2013b). This pilot application concerned a polycyclic soil sequence in driftsand deposits. It showed that the
comparison of pollen and biomarker spectra allowed us to indicate the plant species responsible for carbon
sequestration in the humic horizons (van Mourik & Jansen, 2013b). Important conclusion was that biomarker
analysis showed promise not only in the reconstruction of past local vegetation composition of a specific site,
but also in studies where the emphasis lies not on the vegetation per se, but rather on reconstructing various
sources of soil organic matter input (van Mourik & Jansen, 2013b).
Goal of the present study was to further explore the applicability of biomarker analysis as part of a multi-proxy
reconstruction aimed at unraveling the sources of stable fillings used for the production of plaggic manure in
the context of the historic development of the plaggic agriculture ecosystem. For this, we applied biomarker
analysis on three previous investigated plaggic Anthrosol.
**Materials and methods.**
2.   **Profile selection**

| Fig. 1 |
| --- |


| Fig. 2 a,b,c |
| --- |

The distribution area of plaggic Anthrososl in NW-Europe is indicated in fig.1. Pape (1972) published the first
map of the distribution of plaggic agriculture in NW-Europe. Bastiaens & van Mourik (1995) found traces of
intensivation and extension of this area in Vlaanderen (Belgium) while van Mourik (1999b) also reported
plaggic Anthrosols in Schleswig (Germany). Beside this area with 'real' plaggic Anthrosols, Spek (2004, p. 724)
summarized information about the occurrence of soils with some evidence of application of plaggic manure in
the Atlantic coastal zones of Norway, Denmark, France, Galicia, Scotland and Ireland.
For this pilot study we selected three previously investigated plaggic Anthrosols in the Netherlands with an
undisturbed plaggic horizon: Valenakker, Nabbegat and Posteles (fig.2). Pollen diagrams, [14]C and OSL dates of
these profiles were available and previously published separately  in various articles. Here we combined these,
and re-sampled the plaggic horizons of the profiles for biomarker analysis and new fossil pollen analysis to
allow for comparison. Vertical sampling resolution was: Valenakker 20 cm, Nabbegat and Posteles 10 cm.
Valenakker (van Mourik et al. 2012) is situated southwest of the city Weert (middle Limburg) on the sport fields
of a former college. As a result, during the 20[th] century the soil has never been ploughed or subjected to land
consolidation. This profile has never been affected by roots of *Zea mays*, introduced in The Netherlands in the
middle of the 20[th] century (van Mourik & Horsten, 1995).
Nabbegat (van Mourik et al. 2013a) is situated on the Maashorst (eastern North-Brabant). The plaggic deposits
were buried by drift sand around 1800 AD. Consequently, the plaggic deposits have perfectly been protected
against damage by land consolidation or pollution afterwards (van Mourik et al. 2013a). The site is now
vegetated by oak and birch trees. Roots of these trees may have caused input of organic matter by
decomposed roots in the upper part of the plaggic horizon (fig.3).
| Fig. 3 |
Posteles (van Mourik et.al, 2011) is situated in Twente (eastern Overijssel). The landowner informed us that
during the last three generations this land was never subjected to deep ploughing or land consolidation but
since 1960 *Zea mays* was regular sowed. In contrast to Valenakker and Nabbegat we can expect biomarkers of
this deep rooting cultivated plant.
**2.1. Pollen analysis.**
Pollen diagrams of plaggic Anthrosols provide paleoecological information about plant species, present on site
and in the region during the formation of the plaggic horizon. Previous research showed that pollen grains,
infiltrated in soils and incorporated in plaggic deposits, are well preserved in the anaerobic and acid
microenvironment of excremental aggregates (van Mourik, 1999a, 2001) (fig 4,5).
| Fig. 4 |
| Fig. 5 |
Samples for pollen extraction were collected in 10 ml tubes in profile pits. For a correct matching of pollen and
biomarker spectra of the plaggic deposits, the same samples were treated for both pollen and biomarker
extraction and analysis. The pollen extractions were carried out using the tufa extraction method (Moore et al.,
1991, p. 50). For the identification of pollen grains, the pollen key of Moore et al. (1991, p. 83-166) was applied.
Pollen scores were based on the total pollen sum of arboreal and non-arboreal plant species. For the
estimation of the pollen concentrations of the various soil horizons, the exotic marker grain method was
applied (Moore et al., 1991, p. 53).
**2.2. [14]C and OSL dating**.
The determination of the age of plaggic deposits is subjected to various complications (Spek, 2004). Pollen
stratification is disturbed by bioturbation and ploughing. Besides, the pollen content is a mix of the regular
pollen influx and pollen in stable fillings, used for the production of stable manure (van Mourik et al., 2011).
The ages of humic horizons of buried Podzols cannot be correctly determined by [14]C dating due to the complex
composition of soil organic carbon (van Mourik et al., 1995). During a period of active soil formation, hard
decomposable organic carbon can accumulate in the Ah horizon, especially in the humin fraction but also in the
humic acid fraction. Especially the accumulation of charcoal fragments in the organic aggregates is responsible
for overestimation of the [14]C ages (fig.4). During the Early Holocene small amounts of charcoal fragments were
released after (natural) forest fires, but the amount increased drastic in the iron time due to the charcoal
production for the melting of iron from placic horizons and iron stone (Beukenkamp and Sevink, 2005). The age
of the humic acid fraction was considered the best estimate of the moment of fossilization of the Ah horizon
after burying by driftsand. The difference between humin and humic acids ages was interpreted as a measure
for the period of soil activity and humin accumulation. Later, OSL dating confirmed that radiocarbon dates, not
only of the humin fraction but also of the humic acids, overestimate the true ages (Bokhorts et al., 2005).
Conventional radiocarbon dating of humin and humic acids showed in presented diagrams, extracted from
plaggic deposits, was performed in the CIO (Centre for Isotope Research of the University of Groningen).
OSL dates provide reliable information about the moment of fossilization of plaggic material under the rising
furrow because the quartz grain were perfectly bleached during active ploughing (Bokhorts et al., 2005). OSL
dating of quartz grains, extracted from plaggic deposits, was performed in the NCL (Netherland Centre for
Luminesce Dating, Wageningen University).

**2.3. Biomarker analysis.**
### 2.3.1.    The application of the VERHIB model
A detailed description of the biomarker approach using the VERHIB method is presented in our previous
publications (Jansen et al., 2010; Jansen et al., 2013; Van Mourik & Jansen 2013). Briefly, the basis of the
method lies in the unraveling of the preserved concentration patterns of $C_{20}$-$C_{36}$ $n$-alkanes, which are exclusive
to the epicuticular wax layers on leaves and roots of higher plants (Kolattukudy et al., 1976). While such an
application in itself is not new (e.g. Pancost et al., 2002; Hughen et al., 2004) the novelty of our approach lies in
the application of the  VERHIB model that we specifically developed to unravel the mixed $n$-alkane signal
encountered in soil or sedimentary archives (Jansen et al., 2010). The VERHIB model consists of a linear
regression model that describes how a certain input of plant derived compounds such as $n$-alkanes over time in
a certain archive at a certain location, results in accumulation of these compounds. An inversion of the forward
model is used to reconstruct the accumulation encountered with depth into its most likely vegetation origin
(Jansen et al., 2010). An important aspect of biomarker analysis using VERHIB is that it is an indirect
reconstruction. While the biomarker patterns, in the present study the $n$-alkanes, are directly measured, the
reconstruction into the most likely combination of vegetation biomass input responsible for the observed
pattern is inferred by the model. For this, several parameters must be inputted into the model (Jansen et al.,
2010) the most important of which is the selection of the expected plant species that have been responsible for
the input of biomass in the archive in question, and subsequent inclusion of their $n$-alkane signature in the
VERHIB reference base. In the present study, the selection of species to include was based on the (expected)
crop history of the sites under study, as well as the anticipated origin of the stable fillings used as manure. An
important matter of debate when using $n$-alkane patterns to reconstruct past vegetation input is the genotypic
plasticity of the $n$-alkane patterns, in particular in relation to prevailing environmental factors such as climate
(e.g. Shepherd and Griffiths, 2006). In a previous study focusing on vegetation of relevance for reconstructions
in ecosystems in North-Western Europe where plaggic agriculture occurred, we found that while genotypic
plasticity related to climatic factors may influence the signal, such influence does not eradicate the different
vegetation origins (Kirkels et al., 2013). To limit external influences as much as possible, the vegetation selected
for inclusion in the VERHIB reference base was sampled in close vicinity to the three study sites as much as
possible. The first group of selected plant species concerned the main sources of stable fillings, used for the
manure production: fermented litter from deciduous forest soils (*Quercus robur, Betula pendula*), grass sods
from brook valleys (*Molinia caerulea*) and heath sods (*Calluna vulgaris*).
The second group concerned crop species. Close to the educational Field Study Centre Orvelte (Drenthe) is a
traditional plaggic field where they continued with the cultivation of traditional crop species. There we sampled
*Fagopyrum esculentum, Spergula arvensis, Avena sativa, Secale cereal, Spergula arvensis.* The modern crop
species *Zea mays* corn was sampled on the Posteles.
The concentration patterns of the $n$-alkanes with carbon numbers 20-36 in the selected vegetation samples
and in the soil samples were subsequently used as input for the VERHIB model (see 2.3.2 for a description of
the extraction and analysis of the biomarkers).
A second parameter that must be considered in the application of VERHIB, is input of leaf and root material.
VERHIB considers the species specific $n$-alkane patterns in plant roots separately from the patterns in plant
leaves, and uses this to deal with the input of young root material at depth (Jansen et al., 2010). A first
selection criterion here concerns whether or not leaf and root material can be expected to have entered the
soil at all. For the deciduous forest soil material potentially used as stable fillings (*Quercus robur, Betula*
*pendula*), exclusively leaf derived biomass input is expected as the trees did not grow on-site. In contrast, for
the crop species *Zea Mays* and *Spergula Arvensis* only root material is expected to have entered the soil in
appreciable amounts as the leaf material is mostly removed during harvest. For the other species considered,
both leaf and root material must be taken into account. A selection of root and/or leaf derived n-alkane
patterns to be considered in the VERHIB reference base was made in accordance with the previous. With
respect to the ratio of input of leaf vs. root biomass as required by the model, no exact information is available
for the soil profile under study. Therefore, for those species where both leaf and root material is considered to
have possibly entered the soil, in line with the exploratory nature of the present study, we applied an assumed
leaf/root biomass input ratio of 1.0 and assumed that while input of leaf material always occurred at the top of
the soil profile, root input also occurred with depth. Since our pilot study in polycyclic driftsand deposits
showed that VERHIB was unable to filter out root input sufficiently (Van Mourik & Jansen, 2013), when
interpreting the occurrence of a certain species with depth in the profiles under study as modelled by VERHIB,
the possibility of young root input being responsible for the signal was explicitly taken into account.
Figure 6 shows a flow diagram that illustrates the functioning of the VERHIB modelling as well as the selection
of parameters and reference base species as described above.

| Fig. 6 |
| --- |

### 2.3.2. Extraction and analysis of the biomarkers.
Approximately 0.1 g of each of the freeze-dried and ground vegetation and soil samples was extracted by
accelerated Solvent Extraction (ASE) using a Dionex 200 ASE extractor. The extraction temperature was 75ºC
and the extraction pressure 17 × 106 Pa, employing a heating phase of 5 min and a static extraction time of 20
min. Dichloromethane/methanol (DCM/MeOH) (93:7 v/v) was used as the extractant (Jansen et al., 2006). The
extracts were subsequently fractionated into three fractions containing the *n*-alkanes, the esters and the
combination of alcohols and fatty acids respectively. For this, a silica column consisting of extracted cotton
wool and silica gel was used, followed by elution with hexane, hexane/DCM (4:1) and DCM/Methanol (9:1)
respectively. Separation of the *n*-alkanes took place by on-column injection of 1.0 µl of the first fraction on a 30
m Rtx-5Sil MS column (Restek) with an internal diameter of 0.25 mm and film thickness of 0.1 µm, using He as a
carrier gas. Temperature programming was: 50ºC (hold 2 min); 40ºC/min to 80ºC (hold 2 min); 20ºC/min to
130ºC; 4ºC/min to 350ºC (hold 10 min). Subsequent MS detection in full scan mode used a mass-to-charge
ratio (m/z) of 50-650 with a cycle time of 0.65 s and followed electron impact ionization (70 eV). The *n*-alkanes
were identified from the total ion current (TIC) signal by their mass spectra (dominant fragment ion
represented by m/z = 57) and retention times and quantified using a deuterated internal standard ($d_{42}$-*n*-$C_{20}$
alkane (Jansen et al. 2010) as well as a conventional external *n*-alkane standard.

| Fig.7 |
| --- |

Figure 7 presents the n-alkane biomarker distribution in the leaves and/or roots of the species, inserted in the
reference base. The results show the odd-over-even chain-length predominance typical of higher plants
(Kolattukudy et al., 1976). The observed variation in patterns and concentrations is in line with the variation
found in other species in previous work (e.g. Jansen et al., 2006).

### 3. The vertical distribution of biomarkers and pollen in the analysed profiles.
### 3.1. Profile Valenakker

Profile Valenakker is a plaggic Anthrosol (Aan), overlying a ploughed umbric Podzol (2ABp, 2Bs). The pollen
diagram (fig.3) and the absolute dates (table 1) reflects a soil development of ≈ 1400 year.
The post sedimentary pollen spectra in the 2BS show percentages of tree species as *Corylus* and *Quercus* of the
Middle Subatlantic. The presence of Poaceae, Cyperaceae, *Rumex* and Ranunculaceae reflects a period of
pasture. The high scores of Cerealia in ploughed 2ABp and even the 2B indicate a form of sedentary agriculture
before the start of plaggic agriculture. [14]C dating indicate a carbon age of the base (60 cm) of the Aan horizon
of ≈ 600 AD. The OSL age of the lower part of the plaggic horizon is 800-900 year younger, ≈ 1560 AD.

| Fig.8. |
| --- |

| Table 1. |
| --- |

| Fig.9. |
| --- |

Micromorphological observations (fig.5ab) of the plaggic deposits show the complexity of soil organic matter.
There are various sources of organic carbon as plants roots, tissue of table fillings and sods. Also the
composition of pollen spectra is complex, a mix of the regular pollen influx of plants on the fields and in the
surrounding infiltrating into the soil and pollen, and pollen present in various stable fillings.
In previous studies the origin of stable fillings, used in plaggic agriculture, was reconstructed on the base of
pollen diagrams (Spek, 2004; van Mourik et al., 2012a, 2012b). The pollen spectra of the Aan horizon show very
low scores of arboreal trees but reasonable scores of Ericaceae and Poaceae. Ericaceae pollen may indicate the
use of heath sods, Poaceae pollen the use of grassland sods, the combination of sods from degrading heath and
the rise of the land surface by plaggic manure is caused by the mineral fraction in such sods. However, the rise
of the plaggic horizon of ≈ 60 cm cannot be explained by the use of heath sods if it is true that the use of heath
sods (with a mineral fraction) was introduced in the course of the 18[th] century when better construction
materials enabled the farmers to build deep stables (Vera, 2011). In fact, the sources of stable fillings cannot be
satisfactorily detected with pollen diagrams.
The biomarker spectrum of the base is dominated by *Quercus.* Despite the low percentages *Quercus* pollen it is
very likely that the farmers used forest litter as stable filling. The middle spectrum is dominated by markers of
*Avena* and *Secale*. This points to the use of straw from these crop species as stable filling. Pollen of Cerealia is
present in the whole diagram. In the upper spectrum biomarkers of *Calluna* are present together with *Avena*
and *Secale*. This points to the use of heath sods as additional stable filling during the last phase in the
development of the plaggic horizon.

**3.2. Profile Nabbegat**

| Fig. 10. |
| --- |


| Table 2. |
| --- |


| Fig. 11. |
| --- |

Profile Nabbegat is a haplic Arenosol (with Mormoder humus form), overlying a plaggic Anthrosol, overlying a
ploughed umbric Podzol. The pollen diagram (fig.6.) and the absolute dates (table 2) reflect a soil development
of ≈ 3000 year.
The post sedimentary pollen spectra of the 3ABp reflect the start of agriculture (increase of Cerealia) on a
former heath (decrease of *Ericaceae*) in a surrounding with coppice hedges (*Quercus, Corylus*). Based on
radiocarbon dates, the agricultural activities started before ≈ 1000 BC, the OSL dates point to deposition of
plaggic material after ≈1500 AD.
The radiocarbon ages indicate that the farmers used organic matter with very few mineral 'contamination' for
a long time. The OSL ages indicate that the rise of the plaggic horizon started ≈ 1500 AD due to mineral grains
as part of the manure. The plaggic horizon developed between 1500 and 1800 AD. Around 1800 AD, short after
the introduction of the deep stable economy (Vera, 2011), the plaggic Anthrosol was overblown by driftsand.
Apparently, the use of heath sods resulted in heath degradation, sand drifting and acceleration of the rise of
the plaggic horizon (van Mourik et al., 2012a). The sand drifting stabilized under planted *Quercus* trees; the
roots of these trees reached the buried Anthrosol and may have contributed the scores of biomarkers in the
upperpart of the buried plaggic horizon. The composition of the pollen spectra of the plaggic horizon is rather
uniform, dominated by Ericaceae and Cerealia.
Fig.7. shows the results of biomarker analysis. Biomarkers of *Quercus* were present in all the spectra, dominant
in the lower spectra, regular in the other spectra. This points to the use of forest litter as stable filling during
the development of the lower part of the plaggic horizon. The main crop species during this time was *Spergula*.
The middle part is dominated by markers of *Avena* and *Secale*, indicating the use of straw. Only in the upper
spectrum *Calluna* was found, indicating the use of heath sods during the last phase of the development of the
plaggic horizon.

### 3.3. Profile Posteles

| Fig.12. |
| --- |

| Table 3. |
| --- |

| Fig.13. |
| --- |

Profile Posteles is a plaggic Anthrosol, overlying a ploughed umbric Podzol. The pollen diagram (fig.8) and the absolute dates (table 3) reflect a soil development of at least 1200 year.

The pollen content of the buried ploughed Podzol (2Ap, 2B) is post-sedimentary infiltrated in Late-Glacial coversand by bioturbation and agriculture. Characteristic is the sharp decrease of pollen concentrations with depth, shown by the pollen density curve. The spectra of the 2B horizon already reflect evidence of agriculture (Cerealia) in a deforested landscape (low percentages of *Alnus, Quercus, Fagus*). The spectra of the 2Ap horizon show increasing percentages of Cerealia.

The radiocarbon age of the base of the plaggic deposits (95 cm) is ≈ 850 AD, The OSL age ≈ 1500 AD. The OSL age of the 2Ap (105 cm) is 2035 ± 450 BC, ≈ 3500 year older than sample 95. In this part of the profile we see the effect of bioturbation on the age of the coversand. Grains from the base of the Aan were transported to the 2Ap and reversed, which explains the large standard deviation of the OSL of sample 105 cm.

The actual Ap horizon (the active plough horizon) is palynologically characterized by peak percentages of Cerealia, a slight extension of *Pinus* (planted on the abandoned heath after 1900 AD) and the appearance of *Zea mays* (introduced in Dutch agriculture after 1950 AD). Pollen of Cerealia, Ericaceae and Poaceae were found in all the spectra of the Aan.

The lowest spectrum (80) is dominated by the crop species *Spergula* and the score of *Quercus* indicates the use of forest litter during the development of this part of the Aan.

The spectra 10, 20, 40, 60 are dominated by biomarkers from roots of *Zea mays.* This crop species was introduced around 1950 AD, but the markers of the decomposed *Zea* roots seem to suppress all the others (this was not the case in the profiles Valenakker and Nabbegat). Spectrum 50 is dominated by *Avena* and *Secale*, spectrum 30 by *Zea* and *Secale* and spectrum 0 by *Zea* and *Calluna*. Again the use of heath sods seems restricted to the youngest part of the plaggic horizon.

### 4. Discussion

Pollen diagrams of plaggic Anthrosols provide valuable paleoecological information to reconstruct the soil dynamics during the plaggic agriculture. However, interpretation of pollen diagrams is complicated. Pollen grains, extracted from plaggic deposits, may originate from two sources (van Mourik et al., 2011). The first source concerns the regional pollen influx from flowering species and local flowering crop species. Pollen grains precipitate on the soil surface and may infiltrate into the Anthrosols by ploughing and bioturbation. This pollen influx will be mixed with the pollen content of materials, used as stable filling to produce manure.

Pollen will be preserved in plaggic deposits in the anaerobic and acid micro environment of humic aggregates, produced by worms and micro arthropods (van Mourik, 1999b, 2001). In general it is not possible to make a clear separation between pollen grains originating from the regular pollen influx or from materials as sods. Therefore, the identification of the various sources of stall fillings cannot be based on pollen analysis alone. Additional information, acquired by biomarker analysis proved very useful for this purpose.

In the pollen diagrams *Fagopyrum* is found in almost all the spectra of the plaggic deposits and in Valenakker and Nabbegat even in the top spectra of the buried ploughed Podzol, probably as result of pollen infiltration. *Fagopyrum* as crop species on sandy soils was introduced after 1350 AD (Leenders, 1996). Based on this palynological time marker, plaggic deposition started around 1350 AD.

The radiocarbon ages of plaggic deposits are much older. This is caused by (1) older organic carbon, present in the applied stable fillings (as forest litter) for the manure production and (2) accumulation of hardly decomposable organic carbon during active soil formation. Consequently the radiocarbon dates overestimate the ages of the plaggic sediments, but approach the age of the introduction of agricultural soil management (van Mourik et al., 1995, 2011, 2012a, 2012b). Manuring of infertile soils came already in use in the Bronze Age

and also the Celtic fields are an example of a prehistorical agricultural system based on manure management
(Spek, 2004).
The mineral component of stable manure, applied on the fields, was responsible for the thickening of the
plaggic horizon. Ploughing of the furrow will bleach the OSL signal of the mineral grains until the moment that
the grains are no longer part of the active soil furrow. For that reason, OSL dating of the plaggic horizon provide
reliable ages of the plaggic deposits (Bockhorst et al., 2005). The OSL dates of the profiles Valenakker,
Nabbegat and Posteles indicate a start of the thickening ≈ 1550 AD.
It was not possible to determine the sources of stable fillings palynologically. Possible stable fillings were forest
litter, sods from moist grass lands and heats sods. But in almost all spectra of the pollen diagrams Ericaceae,
Poaceae and arboreal pollen occur. Biomarkers extracted from plaggic deposits, originate from two sources.
The first source concerns biomarkers from decomposed roots of crop species, the second source of organic
material as straw and sods, used as stable filling for manure production.
In the three diagrams we find *Quercus* as dominant marker in the lowest part of the Aan-horizon, indicating the
use of forest litter. In Nabbegat, Quercus markers can also originate from roots of the planted *Quercus* forest
after the stabilization of the sand drifting. This is not the case on Valenakker and Posteles. The middle part of
the Aan-horizon is dominated by markers of *Avena* and *Secale*, indicating the use of straw as stable filling.
Only in the top of the Aan-horizon markers of Calluna are present, indicating  the use of heath sods as stable
filling. Based on the results of the biomarker analysis we can conclude that heaths sods were used as stable
filling only in the 18[th] and 19[th] century. This fits with the observations about the use of heaths in historical
archives Vera (2011).
So the question rises about heath management before the introduction of the deep stable economy. Some
researchers point to careful heath management before the 19[th] century. In interviews with farmers, born
before 1950, Burny (1999) collected essential information about historical heaths management in the Belgian
Kempen. A historical study of land use in the Campina also indicated carefully maintenance and sustainable use
of valuable common fields (de Keyzer, 2014).  Before the 19[th] century, heath sods were never dug on the dry
*Calluna* heath, only on the moist *Erica* heath. These organic sods were not used as stable filling but as fuel for
the furnace. Burning of Calluna heaths was the most important management action to rejuvenate the heath.
Juvenile heath is food for cows. Sods digging was a bad action due to the resistance and incoherence of these
dry sods and also the long recovery period. Mowing of older *Calluna* shrubs took place. Twigs were used for
roofs, burning and brooms.  (Burny, 1999). Because of the very low nutrient contribution to the manure of
mowed *Calluna*, the farmers preferred the use of twigs of broom (*Genista*). When in the course of the 18[th] the
authority relationships changed and the population growth and the demand for food increased, farmers
started to intensify their production (Vera, 2011). They needed more manure and started with the deep stable
economy and the use of *Calluna* heath sods.
An important factor may be the presence of pollen and biomarkers content in sheep droppings. According to
Simpon et al., (1999) biomarkers survive the congestion process and stay in the manure. But what do sheep
consume? Grazing sheep are very selective in collecting food (Oom et al., 2008; Smits & Noordijk, 2013). They
prefer grasses (*Molinia*, *Festuca* and *Corynephorous*). Only in years that there is insufficient grass available at
the end of the summer, they eat shoots of Calluna, at that time nourishing with high concentrations Ca, Mg and
but no P. Pollen extractions from sheep droppings showed that only in droppings, collected during the summer
season *Calluna* pollen is present. During the flowering season of Calluna, the animals consume pollen,
precipitated on the grasses. That explains the presence of Calluna pollen and the absence of Calluna
biomarkers in the lower parts of the plaggic horizons.
If it is true that *Calluna* heath sods were dug only in the 18[th] and 19[th] century, how can we explain the mineral
component in the plaggic manure, responsible of the rise of the land surface  before that time?
According to Smits and Noordijk (2013) there are several sources of minerals. Firstly, a small amount of mineral
grains will be incorporated in the manure during emptying out the manure of the stable. Secondly, farmers had
the knowledge that the addition of sand could improve the fertility of the soil. Not the leached and acid sand
from heath sods but not leached sand, dug on sheep walks and in blown out depressions in nearby drift sand
landscapes.
**5.   Conclusions**
•    The vertical zoning of biomarkers and pollen in plaggic horizons are different. Palynologically, the plaggic
horizon is a homogenous, the biomarker diagrams show differentiation.
•    We can identify various stable fillings used, based on the vertical distribution of biomarkers.
• The biomarker spectra of the base layer of the plaggic horizon are dominated by biomarkers of deciduous
trees litter (dominated by *Quercus*), indicating the use of organic matter from the forest floor.
• The biomarker spectra of the middle part of the plaggic deposits are dominated by crop species (*Avena,*
*Secale*), indicating the use of straw from these species as stable filling during a relatively long time.
• Only the top spectra of the plaggic horizons are dominated *by Calluna,* indicating that heath sods were
used as stable filling only during the last phase in the development of the plaggic horizon.
• Profile Posteles shows the impact of the contribution of biomarkers of roots of *Zea mays*, introduced
around 1950 AD, suppressing the other species.
• The negligible percentages of *Calluna* in biomarker spectra of plaggic deposits with exception of the top,
suggest an overestimating of the use of heath sods in the traditional interpretation of the genesis of
plaggic horizons, the dominance of crop species in biomarker spectra of plaggic deposits suggests
underestimating of the use of straw as source material for the production of organic stable manure to
fertilize ancient arable fields.
**Acknowledgment**
We like to thank Jap Smits (State Forestry) for his information about historical heath management and
agriculture. We are grateful to Annemarie Philip (IBED, University of Amsterdam) for the preparation of the
pollen slides, Hans van der Plicht (CIO, University Groningen) for production of the radiocarbon dates and Jakob
Wallinga (NCL, Wageningen University) for the realization of the OSL dates. The digital illustration were
produced by Jan van Arkel (IBED, University of Amsterdam).

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

## Tables (including table captions)

| Table 1. [14]C and OSL dates of the plaggic deposits of Valenakker. | | | | |
|---|---|---|---|---|
| Horizon | Depth (cm) | Calendric [14]C ages humin | Calendric [14]C ages humic acids | Calendric OSL ages |
| Aan | 20 | – | – | 1775 ± 20 AD |
| Aan | 40 | 771 ± 92 AD | 1049 ± 78 AD | 1635 ± 30 AD |
| Aan | 60 | 595 ± 61 AD | 698 ± 54 AD | 1565 ± 30 AD |

| Table 2. [14]C and OSL dates of the plaggic deposits of Nabbegat. | | | | |
|---|---|---|---|---|
| Horizon | Depth (cm) | Calendric [14]C ages humin | Calendric [14]C ages humic acids | Calendric OSL ages |
| C | 70 | – | – | 1803 ± 12 AD |
| 2An | 80 | 428 ± 107 AD | 626 ± 45 AD | 1770 ± 11 AD |
| 2An | 105 | 37 ± 133 BC | 3 ± 101 AD | – |
| 2An | 130 | 1182 ± 139 BC | 811 ± 101 BC | 1676 ± 14AD |
| 3ABp | 140 | – | 1299 ± 78 BC | – |
| 3ABp | 150 | – | 1385 ±72 BC | – |

| Table 3. [14]C and OSL dates of the plaggic deposits of Posteles. | | | | |
|---|---|---|---|---|
| Horizon | Depth cm | Calendric [14]C ages humin | Calendric [14]C ages Humic acids | Calendric OSL ages |
| Aan | 45 | – | – | 1758 ± 14   AD |
| Aan | 59 | – | – | 1711 ± 20   AD |
| Aan | 70 | 1132 ± 68 AD | 1172 ± 51 AD | 1651 ± 31   AD |
| Aan | 82 | – | – | 1626 ± 20   AD |
| Aan | 95 | 884 ± 82 AD | 861 ± 85 AD | 1517 ± 31   AD |
| 2ABp | 105 | – | – | 2035 ± 450  BC |

**Figure Captions**
Fig. 1.  The location of sampled profiles Valenakker, Nabbegat and Posteles in the distribution area of plaggic
agriculture.
Fig. 2. The plaggic Anthrosols Valenakker, Nabbegat and Posteles. The location of the OSL samples are indicated
in the white circles (depth in cm); the locations of the profiles are indicated in fig. 1.
Fig. 3. Cross-section of a (living) tree root in the thin section of the 2 Aan of Nabbegat (70-80cm). Characteristic
is the double fringing of the root tissue. Such roots were only found in the upper part of the 2Aan of Nabbegat.
Roots of crop species were not found in the thin sections of the three profiles; they decompose rather fast
compared with tree roots.
Fig. 4. Distribution pattern of organic aggregates in a thin section of the Aan of Valenakker (40-50 cm). In the
fabric of the aggregates are charcoal particles visible
Fig. 5.  Pollen grains, visible in a welded aggregate of the same thin sections. Pollen grains in thin sections are
observable as not double fringing, empty spheroidal objects. The palynological characteristics as sculpture and
aperture are not visible without the chemical treatments during pollen extraction.
Fig. 6. Flow diagram of the methodology of biomarker analysis.
Fig. 7. The *n*-alkane biomarker distribution in leaves and/or roots of species sampled, for the reference base of
this pilot study.
Fig. 8. Pollen diagram Valenakker. Pollen density in k.grain/ml.
Fig. 9. Biomarker diagram Valenakker.
Fig. 10. Pollen diagram Nabbegat. Log D = pollen density in log k.grain/ml.
Fig. 11. Biomarker diagram Nabbegat.
Fig. 12. Pollen diagram Posteles; Pollen density in k.grain/ml.
Fig. 13. Biomarker diagram Posteles.

**Figures and tables (including captions)**

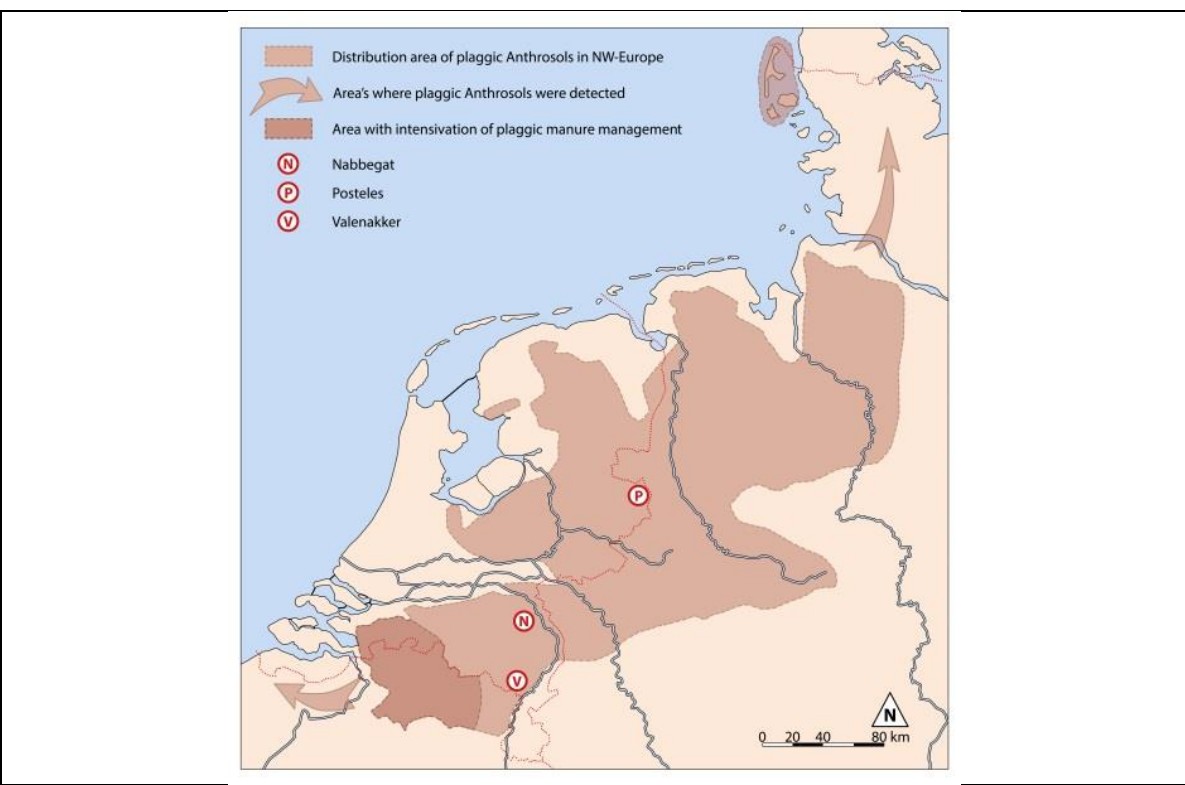

Fig. 1. The location of sampled profiles Valenakker, Nabbegat and Posteles in the distribution area of plaggic agriculture.

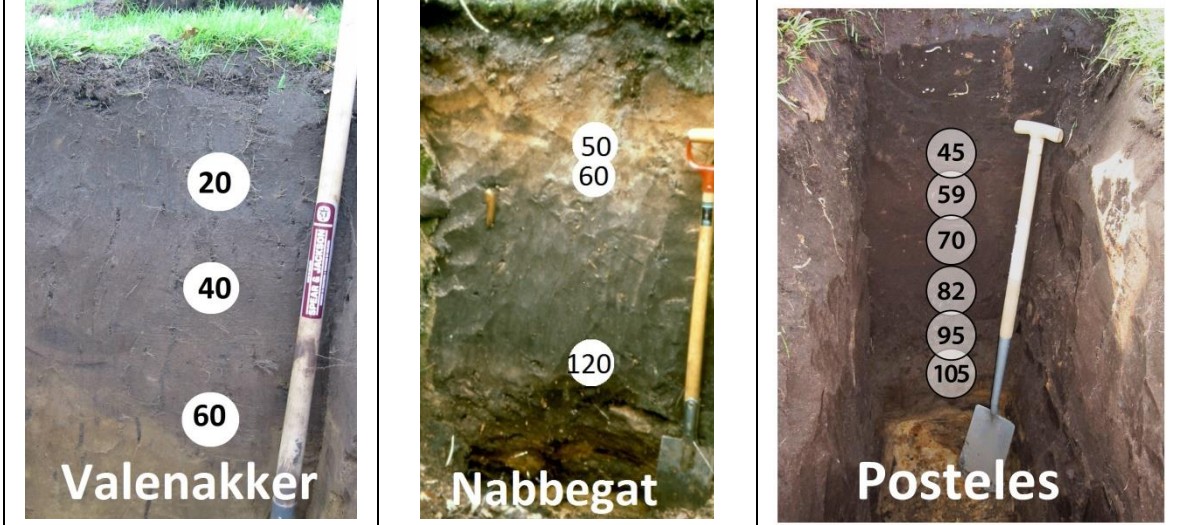

Fig. 2. The plaggic Anthrosols Valenakker, Nabbegat and Posteles. The location of the OSL samples are indicated in the white circles (depth in cm); the locations of the profiles are indicated in fig. 1.

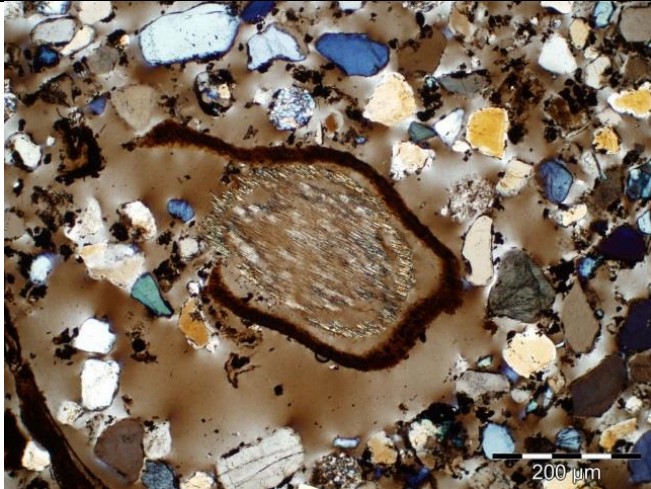

Fig. 3. Cross-section of a tree root in a thin section of the 2 Aan of Nabbegat (70-80cm). Characteristic is the double fringing of the root tissue. Such roots were only found in the upper part of the 2An of Nabbegat. Roots of crop species were not found in thin sections of the three profiles. The decompose rate of such roots is fast.


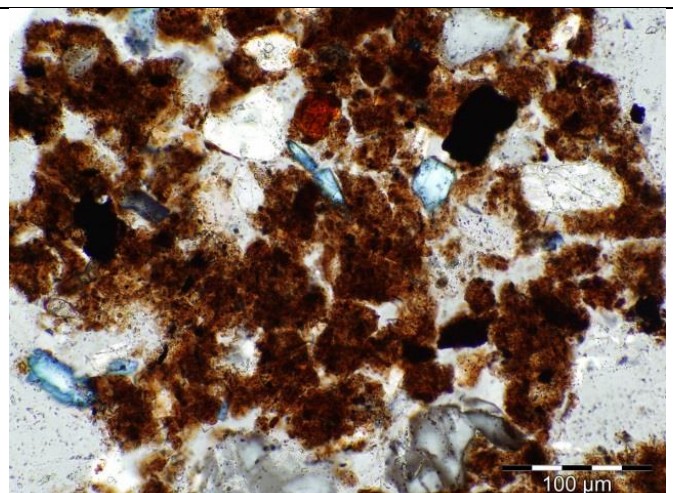

Fig. 4. Distribution pattern of organic aggregates in a thin section of the Aan of Valenakker (40-50 cm). In the intern fabric of the aggregates are charcoal particles visible.


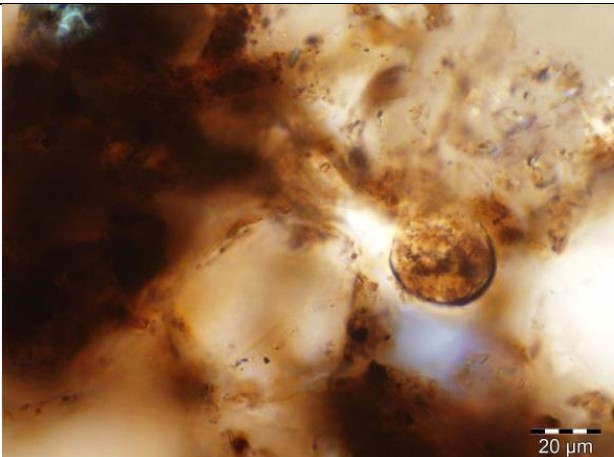

Fig. 5. Pollen grains, visible in an aged aggregate of the same thin sections. Pollen grains are in thin sections observable as not double fringing and empty spheroidal objects. The palynological characteristics as sculpture and aperture are not visible without the chemical treatments during pollen extraction.


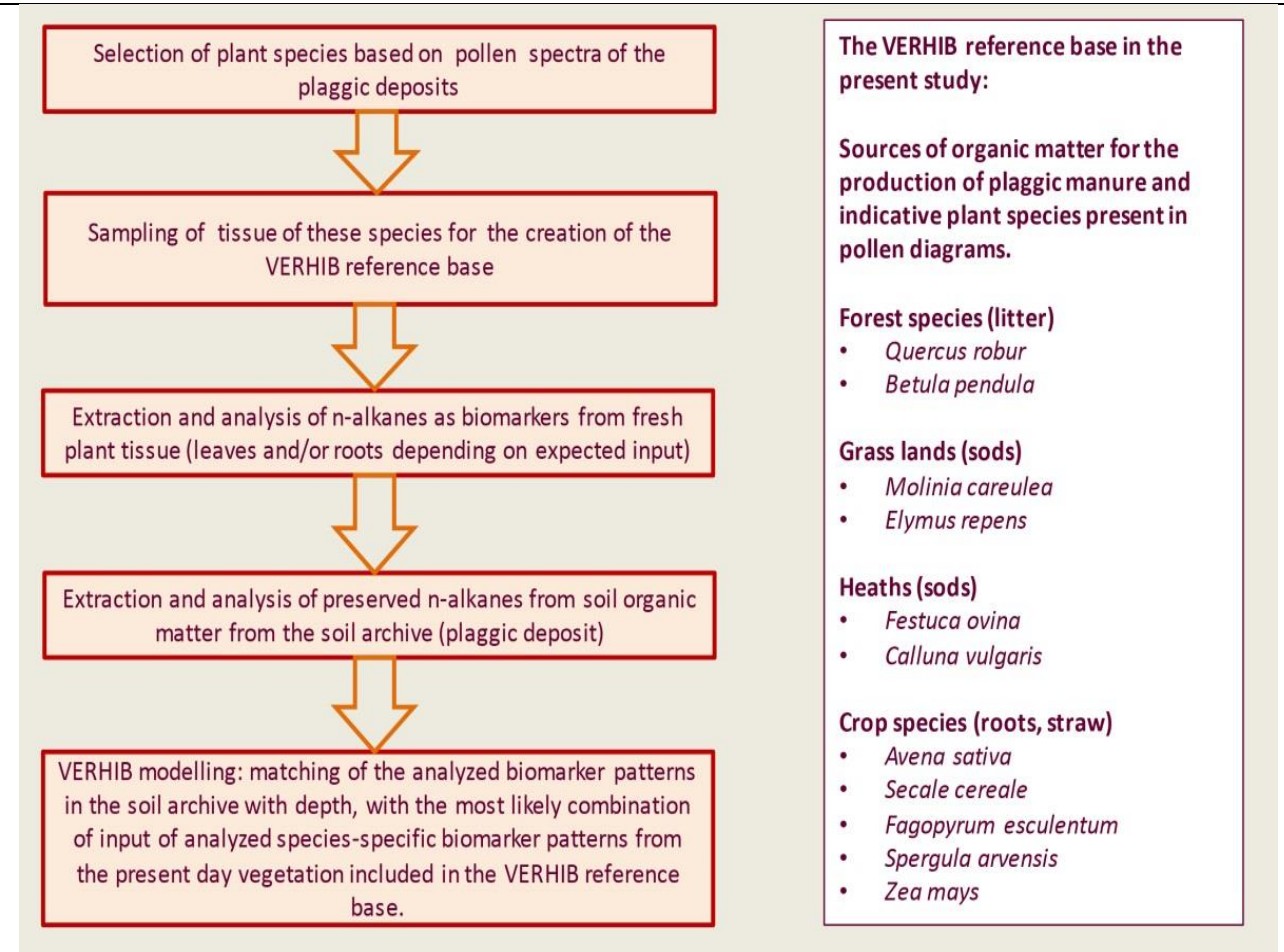

Fig. 6. Flow diagram of the methodology of biomarker analysis.


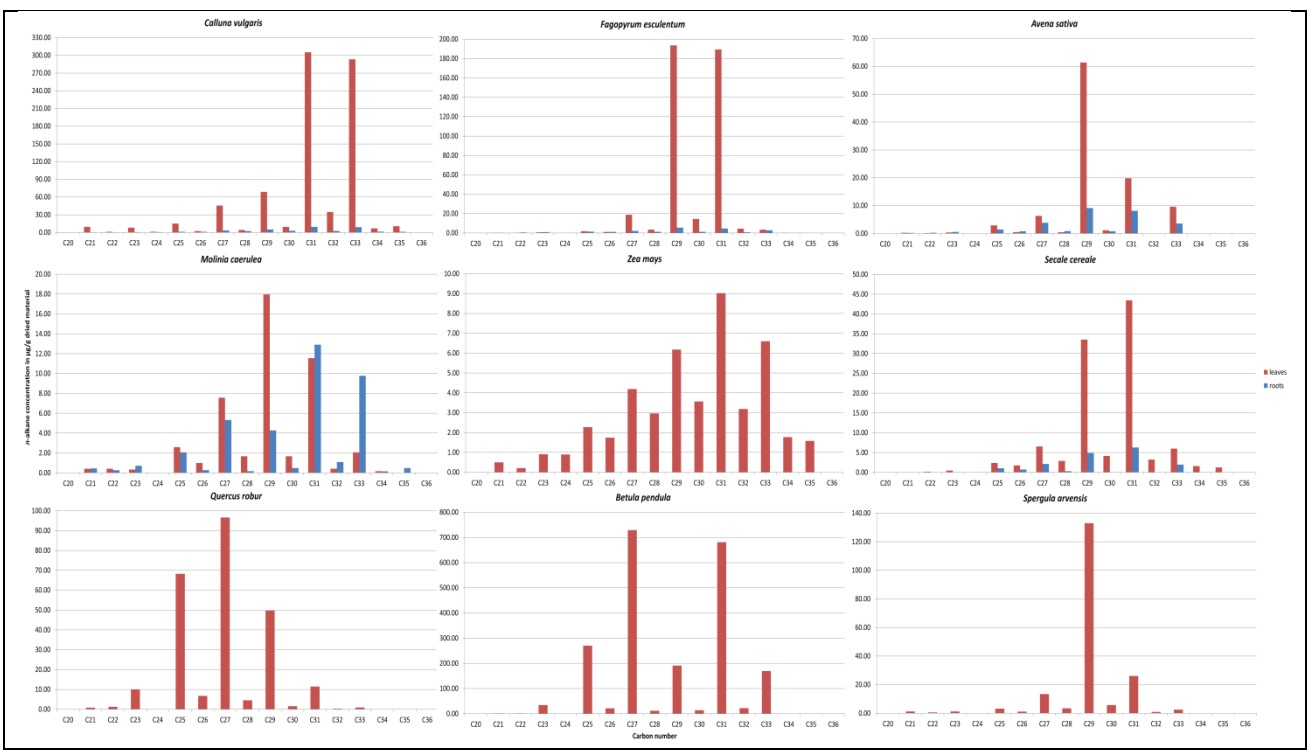

Fig. 7. The *n*-alkane biomarker distribution in leaves and/or roots of species sampled, for the reference base of this pilot study.


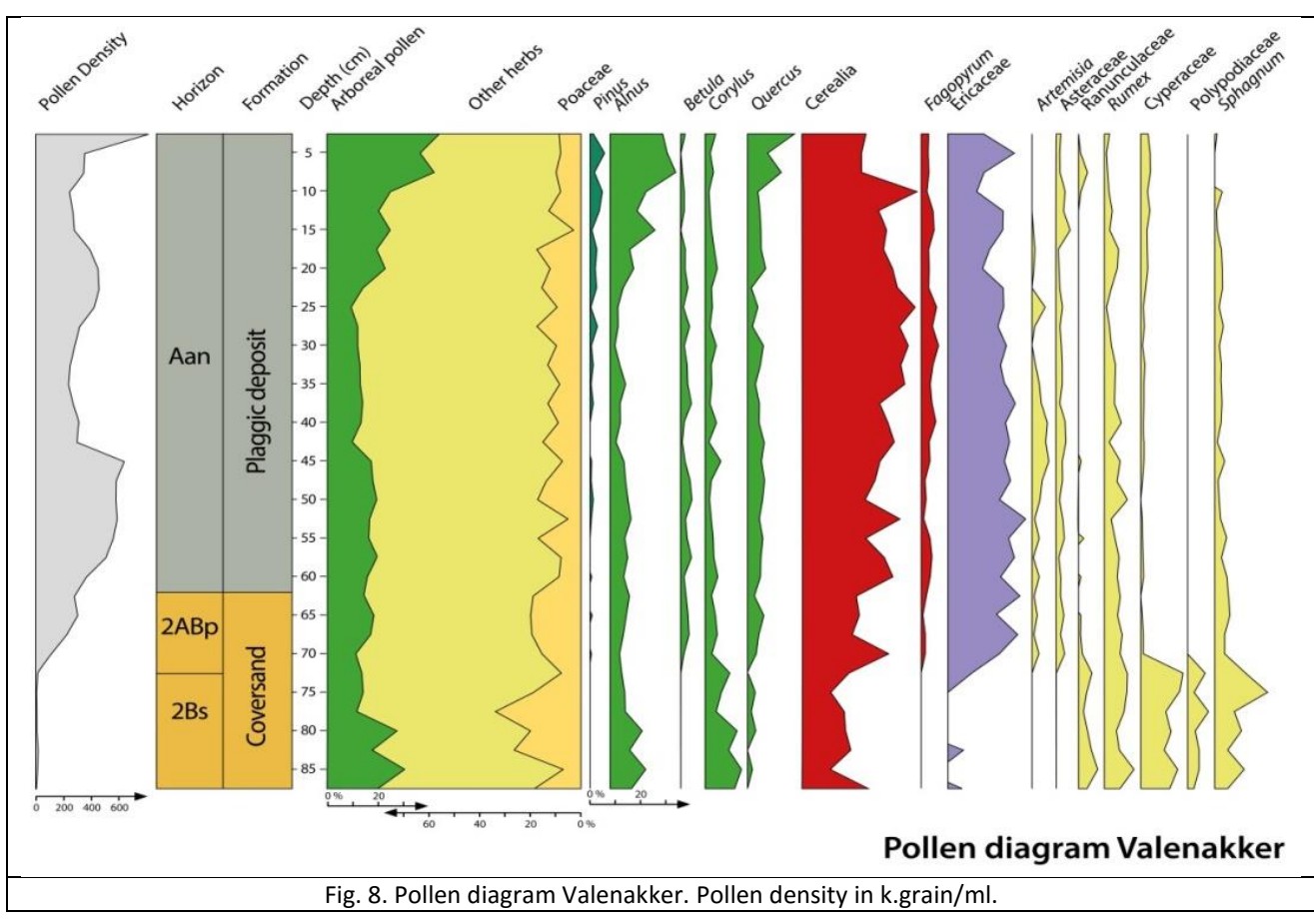

Fig. 8. Pollen diagram Valenakker. Pollen density in k.grain/ml.


| Table 1. [14]C and OSL dates of the plaggic deposits of Valenakker. | | | | |
|---|---|---|---|---|
| Horizon | Depth (cm) | Calendric [14]C ages humin | Calendric [14]C ages humic acids | Calendric OSL ages |
| Aan | 20 | – | – | 1775 ± 20 AD |
| Aan | 40 | 771 ± 92 AD | 1049 ± 78 AD | 1635 ± 30 AD |
| Aan | 60 | 595 ± 61 AD | 698 ± 54 AD | 1565 ± 30 AD |


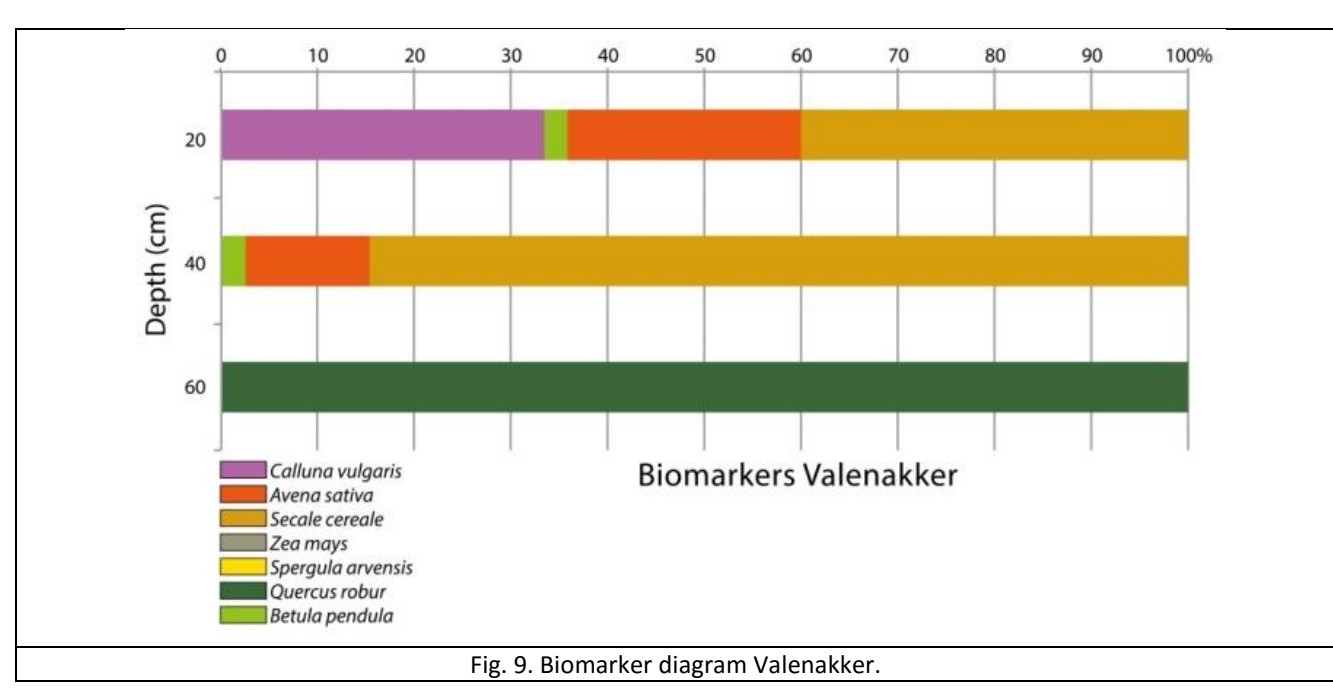

Fig. 9. Biomarker diagram Valenakker.


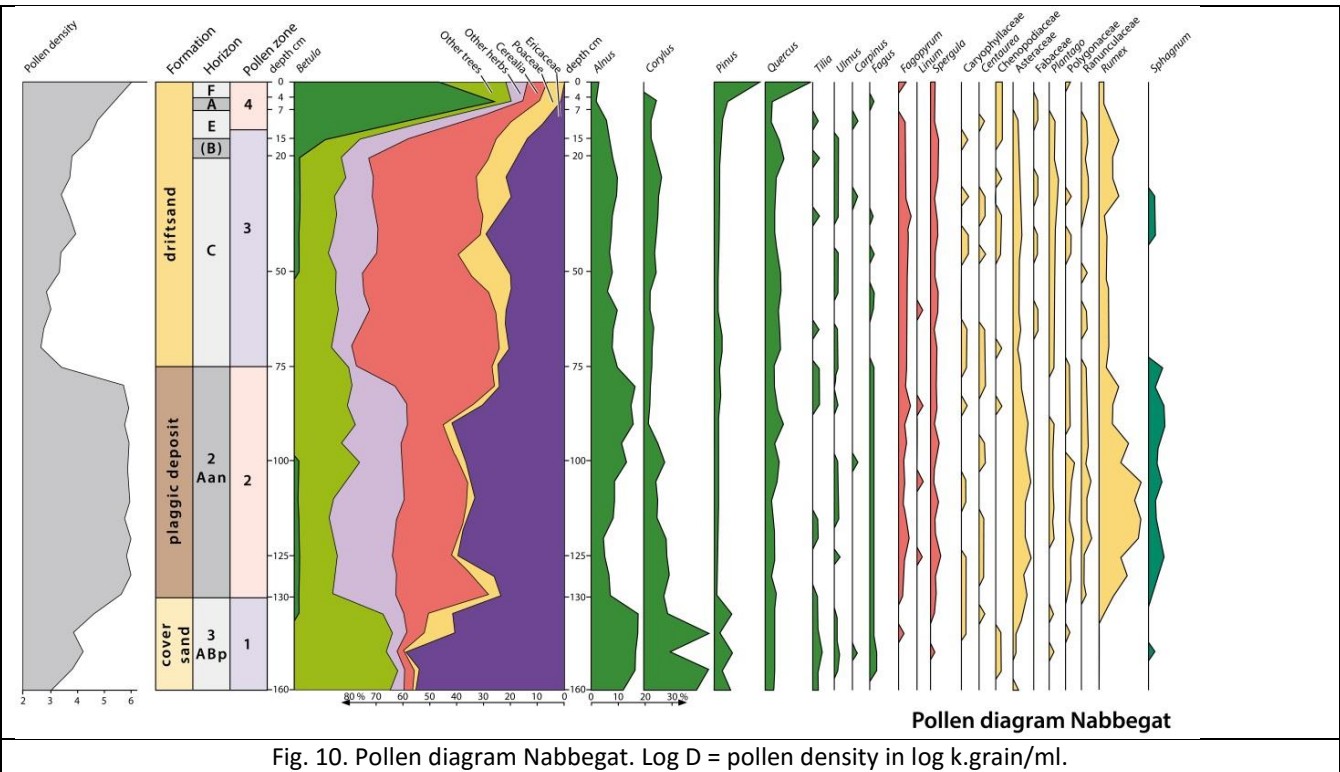

Fig. 10. Pollen diagram Nabbegat. Log D = pollen density in log k.grain/ml.

| Table 2. ¹⁴C and OSL dates of the plaggic deposits of Nabbegat. | | | | |
|---|---|---|---|---|
| Horizon | Depth (cm) | Calendric ¹⁴C ages humin | Calendric ¹⁴C ages humic acids | Calendric OSL ages |
| C | 70 | – | – | 1803 ± 12 AD |
| 2An | 80 | 428 ± 107 AD | 626 ± 45 AD | 1770 ± 11 AD |
| 2An | 105 | 37 ± 133 BC | 3 ± 101 AD | – |
| 2An | 130 | 1182 ± 139 BC | 811 ± 101 BC | 1676 ± 14AD |
| 3ABp | 140 | – | 1299 ± 78 BC | – |
| 3ABp | 150 | – | 1385 ±72 BC | – |

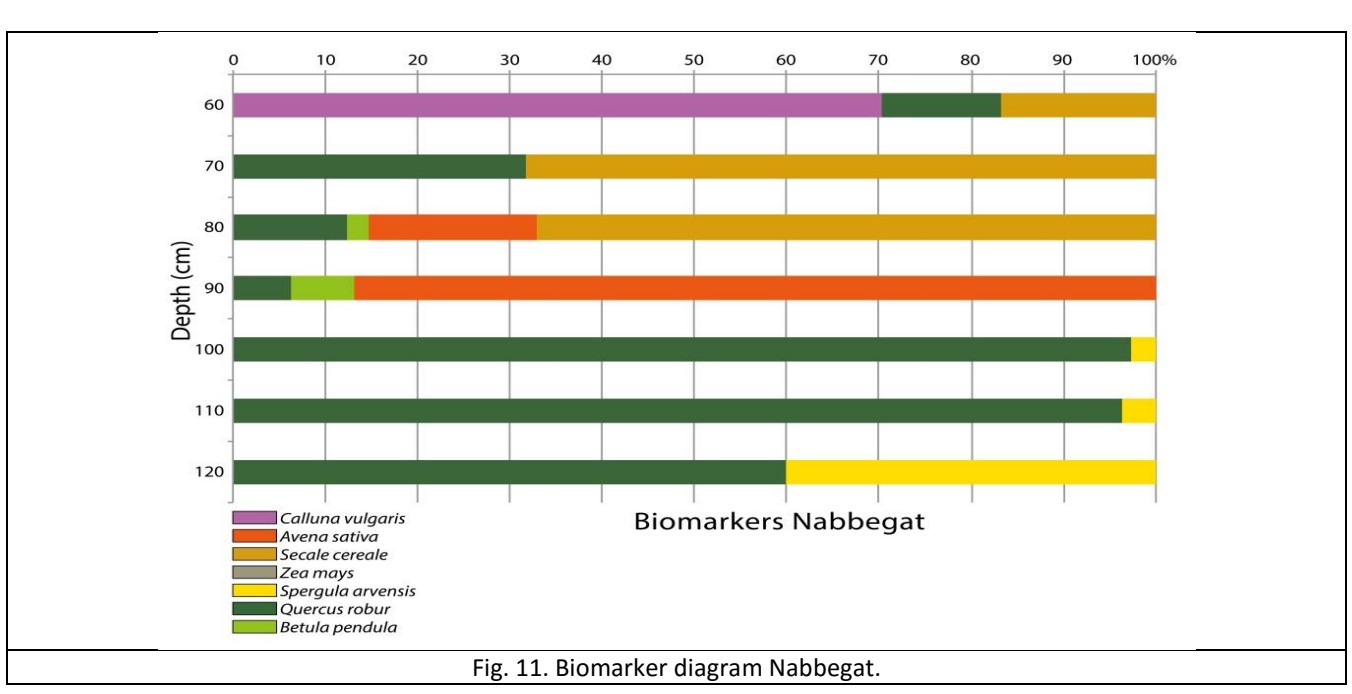

Fig. 11. Biomarker diagram Nabbegat.

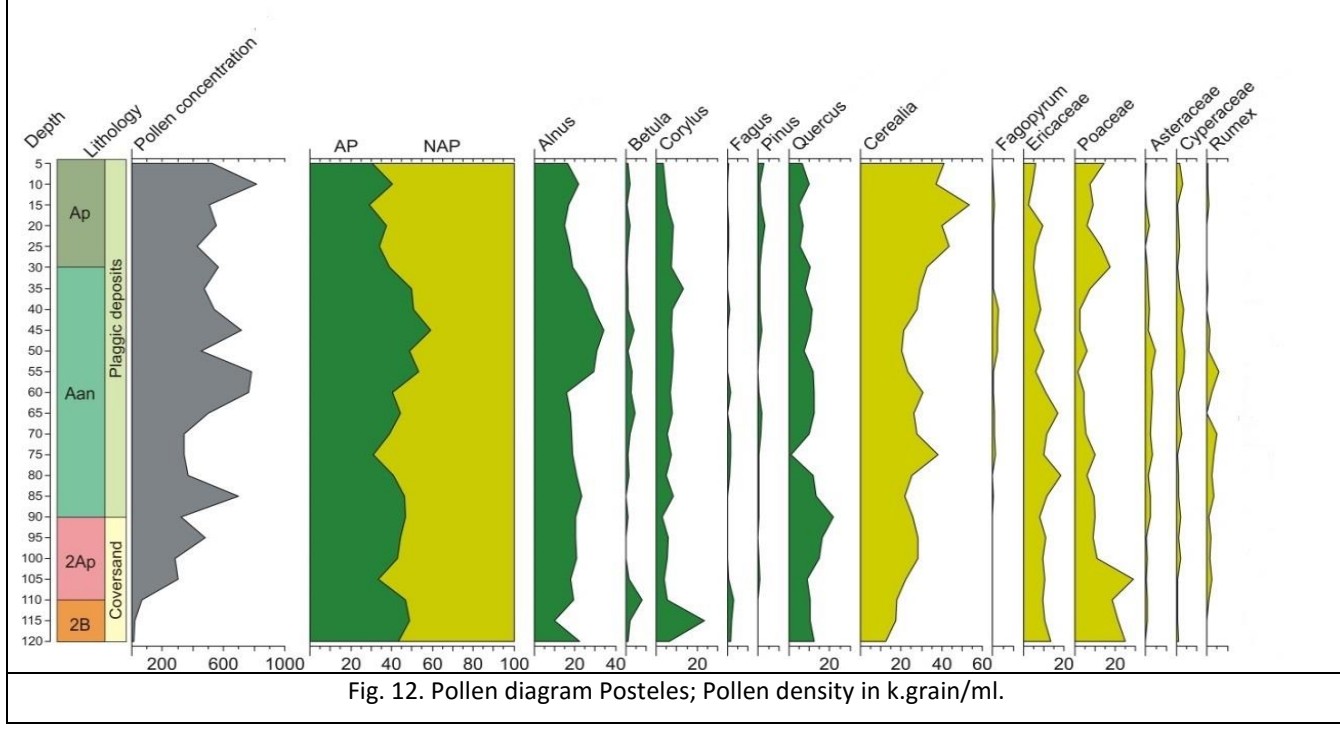

Fig. 12. Pollen diagram Posteles; Pollen density in k.grain/ml.


| Table 3. [14]C and OSL dates of the plaggic deposits of Posteles. | | | | |
|---|---|---|---|---|
| Horizon | Depth cm | Calendric [14]C ages humin | Calendric [14]C ages Humic acids | Calendric OSL ages |
| Aan | 45 | – | – | 1758 ± 14   AD |
| Aan | 59 | – | – | 1711 ± 20   AD |
| Aan | 70 | 1132 ± 68 AD | 1172 ± 51 AD | 1651 ± 31   AD |
| Aan | 82 | – | – | 1626 ± 20   AD |
| Aan | 95 | 884 ± 82 AD | 861 ± 85 AD | 1517 ± 31   AD |
| 2ABp | 105 | – | – | 2035 ± 450  BC |


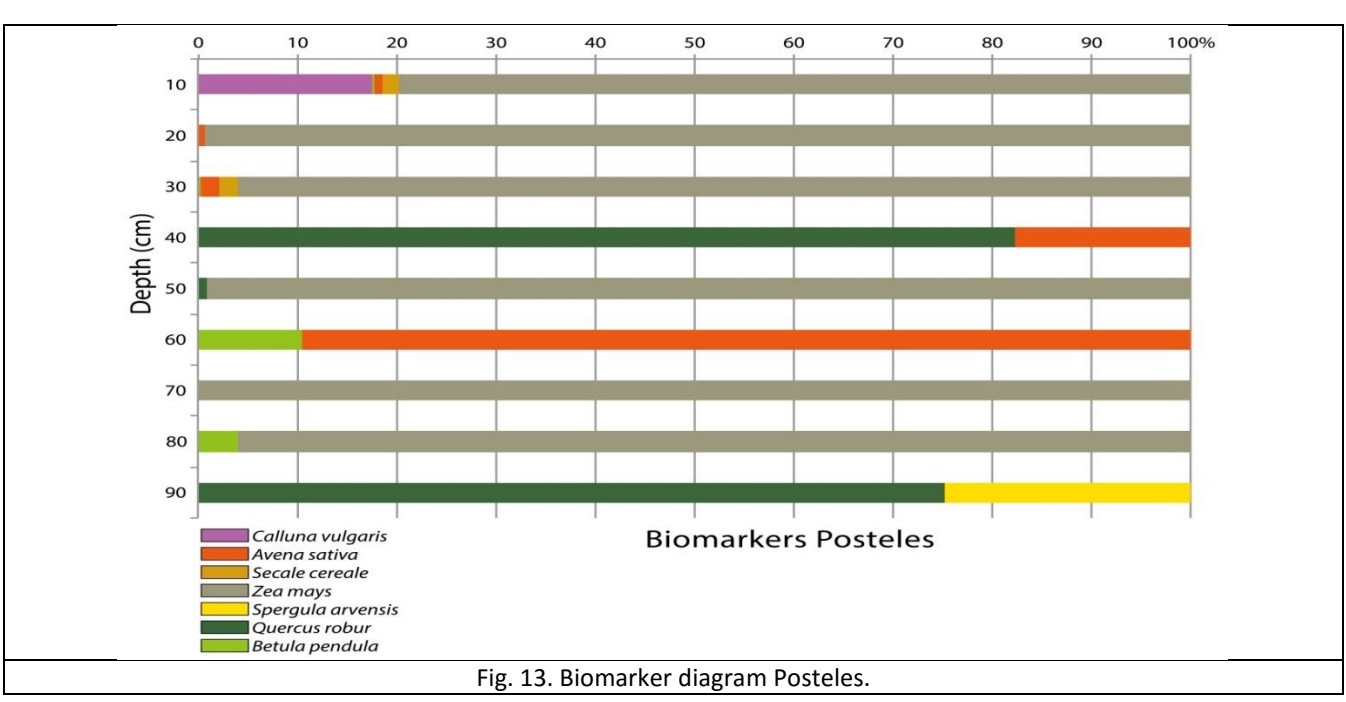

Fig. 13. Biomarker diagram Posteles.

**Figures 1 – 13 (including figure captions).**