# Peer review of "The added value of biomarker analysis to the genesis of Plaggic Anthrosols; the identification of stable fillings used for the production of plaggic manure."

_SOIL, 2015_

## Referee Comment (RC1) · J. Wallinga (Referee) · 11 Feb 2016

**Van Mourik et al. – The added value of biomarker analysis to the genesis of Plaggic Anthrosols – the identification of stable fillings used for the production of plaggic manure.**

**Submitted to SOILD; reviewed by Jakob Wallinga (jakob.wallinga@wur.nl)**

The authors investigate to what extent biomarker analysis can contribute to identify what organic plant material was used as stable filling and contributed to the development of Plaggic Anthrosols, and how the land surface rise at early stages of plaggic agriculture can be explained. The study is based on a preliminary investigation at three sites in the Netherlands, and builds on existing information with regards to the chronology of the plaggic soil development, as well as the micro-morphology and pollen content of the soils.

The manuscript reads smoothly, and provides some interesting new insights into the development of the Plaggic Anthrosols. However, there are a few key issues that will need to be modified prior to publication:
1) The manuscript contains a number of serious mistakes that need to be corrected.
2) It should be made much clearer which part of the information (e.g. figs.) is original, and which parts were published before.
3) Emphasis should be on the novel aspects of the work (the biomarkers), and this aspect should be discussed in more detail.

Each of these issues is detailed below, and should be corrected before publication can be considered. I note that there is considerable overlap with earlier publications; whether this is acceptable I leave to the editor.

**1) Essential corrections**

The two thin sections from plaggic horizons of Valenakker shown in Figure 5 have been published before (Van Mourik et al., 2012 – QI) and were attributed to different locations in that original work. Fig. 5a was published as Fig. 12 in the original work (Van Mourik et al., 2012 – QI) where it was reported to be from Slabroek (*Fig. 12. Micro photo of the distribution of intertextic organic aggregates in the 2Ah horizon, profile Slabroek. The distribution pattern of the aggregates is in line with the regular pattern, observed in Ah horizons of acid podzols......').* Fig. 5b was published as Fig. 15 in the original work (Van Mourik et al., 2012 – QI)), where it was reported to be from Nabbegat (*Fig. 15. Micro photo of the internal fabric of an organic aggregate in the 2Aan horizon profile Nabbegat with (in the centre of the photo) a pollen grain and a charcoal fragment.'* In addition, these same photos were used by Van Mourik et al. (2012-Intech – Fig. 31 and 32), where they were attributed to Valenakker. At least one of the attributions in existing literature and/or the present manuscript must be wrong, and corrections should be made.

In addition, OSL ages in Table 1 (Valenakker) and 2 (Nabbegat) are identical; those given in Table 2 are incorrect and differ from those presented for Nabbegat in the original work (Van Mourik et al., 2012 – QI, Table 7). The C-14 ages are also different to those published before, and the sample depth for the geochronological samples seems to be incorrect.

Table 6 from Van Mourik et al., 2012 – QI:

**Table 7**
$^{14}$C and OSL ages of profile Nabbegat.

| Horizon | Depth (m) | OSL age | $^{14}$C age hac | $^{14}$C age hum |
|---------|-----------|---------|------------------|------------------|
| C | 0.70 | 1803 ± 12 AD | – | – |
| 2An | 0.80 | 1770 ± 11 AD | 580–670 AD | 400–610 AD |
| 2An | 1.05 | – | 90 BC–130 AD | 177 BC–120 AD |
| 2An | 1.30 | 1676 ± 14 AD | 960–760 BC | 1370–1040 BC |
| 3ABp | 1.40 | – | 1390–1220 BC | – |
| 3ABp | 1.50 | – | 1490–1310 BC | – |

Finally, the text and figures are not always in agreement; where Fig. 9 indicates dominance of Zea mays at 10, 20, 30, 50, 70 and 80 cm at Posteles, the tex (p. 13) indicates dominance at 10, 20, 40 and 60 cm. Also the other attributes do not link.......

**2) Originality material**

All information in the Tables and the majority of the figures have been published before. The information is important for this paper, and hence there is good reason to present the information here. However, it should be made clear that the information is not original (in the text, but also in the figure caption). In addition, the editor should decide whether the amount of recycling is acceptable and whether the amount of new information is sufficient to warrant publication.

Fig. 1 – is very similar to Bokhorst et al. 2005; which states the original is from Pape
Fig. 2 – a) has been published in 2012 intech, b) in 2012 QI, and c) in 2011 Catena
Fig. 3 – has been published in Van Mourik et al., 2012 – Intech (and possibly before?)
Fig. 4 – NEW
Fig. 5 – identically (incl. caption) published before in Van Mourik et al., 2012 – intech; photo's published as well in Van Mourik et al., 2012 – QI (with different allocations, see above).
Fig. 6 – published before in Van Mourik et al., 2012 – QI (and there cited to be from Van Mourik and Ligtenberg, 1988)
Fig. 7 – NEW
Fig. 8 – published before in Van Mourik et al., 2011 – Catena (and possibly before?)
Fig. 9 – NEW

**3) Novelty**

The novelty of the work clearly relates to the investigation of biomarkers in plaggic deposits. This aspect should be discussed in far more detail:

- An input of leaf/root biomass ratio of 1.0 is applied 'in line with the exploratory nature of the present study'. How important is this ratio? What would happen if a different ratio is assumed (e.g. ½ or 2?). In the results, I don't see a distinction between root and leaf being made; is the ratio relevant at all to this work? In a previous paper by Van Mourik & Jansen (2013), they state that differences between pollen and biomarker presence can help interpretation: *'c) Biomarkers indicate the presence of a species at a certain interval, but pollens do not. This implies that the species was present, but not in this particular interval. Instead the biomarkers originate from younger root input from a higher (younger) soil surface of the sequence.'* Was such an approach attempted here, and if not, why not?
- The most detailed biomarker profile (Fig. 9; Posteles) shows huge fluctuations, which are not at all discussed in the text. How can biomarkers of Zea Mays completely dominate the spectra at 30, 50 and 70 cm depth, but not be present at all at 40 and 60 centimetres. Were the samples too small to be representative ? What implications does this have for interpretation of the other profiles?
- How about decomposition of biomarkers; is it possible that Calluna vulgaris biomarkers degrade over time, and therefore are not present at depth?

In the final paragraphs of the discussion, the question is raised how the mineral component in the plaggic manure, responsible for the rise of the land surface at the earlier stages can be explained. This indeed is an interesting question, that should be discussed in more detail. The idea that the minerals may be from the stables (the floor?), is interesting. However, I presume that the area of the stables is much smaller than the area of the raised fields (1:100?). If so, raising the field by one cm of mineral material would require the stable floor to be lowered by a metre..... The second option (unleached sand dug on sheep walks and blown out depressions) is interesting; is there any evidence? Could you calculate how much material would be needed, and is the area of sheep walks sufficient to provide the material needed (similar to the argument above). Finally, could there be a direct input from drifting sand as well? Please expand this discussion.

**References:**
Bokhorst, M. P., Duller, G. A. T., and van Mourik, J. M.: Optically Stimulated Luminescence Dating of a fimic anthrosol in the Southern Netherlands, J. Archaeol. Sci., 2005, 547–553, 2005.

van Mourik, 5 J. M. and Jansen, B.. The added value of biomarker analysis in palaeopedology; reconstruction of the vegetation during stable periods in a polycyclic driftsand sequence in SE-Netherlands, Quat. Int. 306, 14–23, 2013

van Mourik, J. M., Slotboom, R. T., andWallinga, J.: Chronology of plaggic deposits; palynology, radiocarbon and optically stimulated luminescence dating of the Posteles (NE-Netherlands), Catena, 84, 54–60, 2011.

van Mourik, J. M., Seijmonsbergen, A. C., and Jansen, B.: Geochronology of Soils and Landforms in Cultural Landscapes on Aeolian Sandy Substrates, Based on Radiocarbon and 15 Optically Stimulated Luminescence Dating (Weert, SE-Netherlands), InTech (2012), Radiometric Dating, 75–114, 2012.

van Mourik, J. M., Seijmonsbergen, A. C., Slotboom, R. T., and Wallinga, J.: The impact of human land use on soils and landforms in cultural landscapes on aeolian sandy substrates (Maashorst, SE Netherlands), Quat. Int., 265, 74–89, 2013.

In addition; the recent thesis of Maika de Keyzer (Antwerpen University, Belgium) contains interesting information on the farming system in the Kempen. This work could/should be referenced here.

---

## Referee Comment (RC2) · Anonymous Referee #2 · 18 Feb 2016

General Comment

Van Mourik et al. investigated Plaggic Anthrosols using pollen and n-alkane biomarkers in order to identify stable fillings that were used for the production of plaggic manure. The presented manuscript is basically well written and the presented study is worth publication. However, given the considerable overlap with already published papers, the manuscript will profit from a stronger focus on the novel biomarker results.

Particularly, the alkane patterns of the vegetation samples are the basis for the interpretation of the biomarker patterns of the soil profiles. However, respective results

for vegetation (above ground and root samples) are not presented. Is it possible to visualize the VERHIB method somehow?

Furthermore, while genotypic plasticity of alkane patterns is discussed, aspects like possible degradation effects and different alkane production/concentration of plants/roots are missing in the current version of the manuscript.

I strongly encourage the authors to perform compound-specific $\delta$13C analyses on some of their alkane samples from profile Posteles. According to the authors' VER-HIB method, several soil samples contain nearly 100% alkanes from Zea mays roots. Given that Zea mays is a C4 plant, this would result in a clear $\delta$13C signal of the alkanes and would thus be as very strong cross check validating (and thus convincing the readers) or falsifying the approach of the authors.

Specific comments

Subchapter 2.2 14C and OSL dating: This was not carried out within this study but is already published. I therefore suggest to delete this subchapter and to incorporate the information in the text that is preceding subchapter 2.1 Pollen. Furthermore, include source/citations in Tables 1-3.

p.10, l.18: I find this formulation inappropriate, because the radiocarbon age does not reflect that the Anthrosol soil development started/lasted 1400 years (ago). The same holds true for the identical formulation in the subsequent subchapters.

p.10, l.20: What do you mean with "post sedimentary pollen spectra"?

p.11, ll.1-4: Micromorphology is not listed in the Material & Method section. Either include there, or, if already published, then delete here in the Result chapter and include with citation in the Discussion chapter.

p.13, ll.17-19: You fail to tell your readers that you refer to biomarkers here and you fail to refer to the respective figure. Furthermore, please check, 80 is not the lowest spectrum, is it?

p.13, l.20: If you can differentiate between leaf and root-derived alkane biomarkers, please show these results also in your figures showing the biomarker spectra. I would appreciate to see the respective alkane patterns and concentrations of your root samples.

p.14, l.24: This is not clear to me. Why should accumulation of organic matter result in a radiocarbon age overestimation?

p.14, ll.27 - p.15, l.2: Is this derived from radiocarbon results? If yes, this statement is not robust/correct due to the age overestimation

p.15, l.13: Evidence for root-derived alkanes is not traceable based on the presented results.

p.18, ll.15-17: delete, not listed in the text

p.20, ll. 23-25: delete or include in the text

---

## Author Comment (AC1) · 1 Apr 2016

"The added value of biomarker analysis to the genesis of Plaggic Anthrosols – the identification of stable fillings used for the production of plaggic manure." J. M. van Mourik, T. Wagner, J. G. de Boer, and B. Jansen

We thank the reviewers fort their carefully reviews and the critical comments to improve the quality of this paper.

It is correct that we used the pollen diagrams and dating result of previous publications, but in this paper we bring the results of 3 plaggic Anthrosols together, to show the

potential value of the new analytical technique of lipid biomarker analysis when used in combination with existing approaches. We agree with the reviewers that this should be made clearer in the manuscript, and that more emphasis should be placed on the new aspects as provided by lipid biomarker analysis.

Our intention is to: 1. More explicitly indicate which data is new, and which is based on previous work. 2. Replace the earlier used micro photographs by new photographs (from the existing thin sections). 3. Present more extensive information about the functioning of the lipid biomarker approach, specifically indicating how certain parameters (e.g. root vs. leaf input) are selected; and support this by a new figure with a flow diagram giving a systematic overview of the process of biomarker analysis and modelling. 4. Present and discuss the biomarker data more extensively, including new figures showing lipid concentration patterns of all relevant plant species and tissues (new data not shown now). 5. Carefully follow and implement the other recommendations / corrections as suggested by the reviewers.
* * *

---

## Author Response (AR2)

Point-by-point response to the reviews, a list of relevant changes.

1.   Additional text for biomarker paragraph and two figures (6, flow diagram and 7, mass spectra
   for reference base) to explain better the biomarker analysis procedure.
2.   We inserted two sentences (incl. references) to inform better about heath management and
   the consequences for species in pollen and biomarker spectra.
3.   We corrected the tables of the [14]C and OSL dates and the depths of some spectra.
4.   We removed the two microphotographs and replaced them by 3 brand new pictures (figures
   3, 4 and 5).

[revised manuscript text omitted]